# Comprehensive Analysis of CXCR4, JUNB, and PD-L1 Expression in Circulating Tumor Cells (CTCs) from Prostate Cancer Patients

**DOI:** 10.3390/cells13090782

**Published:** 2024-05-03

**Authors:** Argyro Roumeliotou, Areti Strati, Foteini Chamchougia, Anastasia Xagara, Victoria Tserpeli, Stavroula Smilkou, Elina Lagopodi, Athina Christopoulou, Emmanouil Kontopodis, Ioannis Drositis, Nikolaos Androulakis, Vassilis Georgoulias, Filippos Koinis, Athanasios Kotsakis, Evi Lianidou, Galatea Kallergi

**Affiliations:** 1Laboratory of Biochemistry/Metastatic Signaling, Department of Biology, University of Patras, 26504 Patras, Greece; argyroumi@gmail.com (A.R.); up1073793@ac.upatras.gr (F.C.); 2Analysis of Circulating Tumor Cells Lab, Laboratory of Analytical Chemistry, Department of Chemistry, National and Kapodistrian University of Athens, 15771 Athens, Greece; astrati@chem.uoa.gr (A.S.); victoriatserp@gmail.com (V.T.); stavroulasmilkou2395@gmail.com (S.S.); elinalagopodi94@gmail.com (E.L.); lianidou@chem.uoa.gr (E.L.); 3Faculty of Medicine, School of Health Sciences, University of Thessaly, 41500 Larissa, Greece; xagaraa@hotmail.com (A.X.); fkoinis@uth.gr (F.K.); thankotsakis@uth.gr (A.K.); 4Hellenic Oncology Research Group, 11526 Athens, Greece; georgsec@med.uoc.gr; 5Oncology Unit, ST Andrews General Hospital of Patras, 26332 Patras, Greece; athinachristo@hotmail.com; 6Department of Oncology, Venizeleion General Hospital of Heraklion, 71409 Heraklion, Greece; kontopodise@gmail.com (E.K.); drositis@venizeleio.gr (I.D.); nandroulakis@yahoo.gr (N.A.); 7First Department of Medical Oncology, Metropolitan General Hospital, 15562 Athens, Greece

**Keywords:** circulating tumor cells, mRNA, prostate cancer, CXCR4, JUNB, PD-L1

## Abstract

CXCR4, JUNB and PD-L1 are implicated in cancer progression and metastasis. The current study investigated these biomarkers in CTCs isolated from metastatic prostate cancer (mPCa) patients at the RNA and protein levels. CTCs were isolated from 48 mPCa patients using the Ficoll density gradient and ISET system (17 out of 48). The (CK/PD-L1/CD45) and (CK/CXCR4/JUNB) phenotypes were identified using two triple immunofluorescence stainings followed by VyCAP platform analysis. Molecular analysis was conducted with an EpCAM-dependent method for 25/48 patients. *CK-8*, *CK-18*, *CK-19*, *JUNB*, *CXCR4*, *PD-L1*, and *B2M* (reference gene) were analyzed with RT-qPCR. The (CK+/PD-L1+/CD45-) and the (CK+/CXCR4+/JUNB+) were the most frequent phenotypes (61.1% and 62.5%, respectively). Furthermore, the (CK+/CXCR4+/JUNB-) phenotype was correlated with poorer progression-free survival [(PFS), HR: 2.5, *p* = 0.049], while the (CK+/PD-L1+/CD45-) phenotype was linked to decreased overall survival [(OS), HR: 262.7, *p* = 0.007]. Molecular analysis revealed that 76.0% of the samples were positive for *CK-8,18*, and *19*, while 28.0% were positive for *JUNB*, 44.0% for *CXCR4*, and 48.0% for *PD-L1*. Conclusively, CXCR4, JUNB, and PD-L1 were highly expressed in CTCs from mPCa patients. The CXCR4 protein expression was associated with poorer PFS, while PD-L1 was correlated with decreased OS, providing new biomarkers with potential clinical relevance.

## 1. Introduction

Prostate cancer is the second most common cancer and the fifth leading cause of cancer mortality in men worldwide [1]. Newly diagnosed patients are usually treated with the standard treatment approach of androgen deprivation therapy, although in many cases the development of resistance leads to metastasis [2]. Prostate cancer patients with distant metastases, particularly to the bone, have limited effective treatment options and poorer disease outcomes [2,3]. Therefore, early diagnosis combined with the discovery of new treatment approaches is crucial for effective treatment of even the most aggressive forms of prostate cancer [4].

Liquid biopsy is an effective and minimally invasive tool for discovering new therapeutic targets and monitoring their efficacy in real time. Tumor circulome in liquid biopsies, unlike tissue biopsies, can elucidate the clonal variation of tumors [5]. Liquid biopsy components which include circulating tumor cells (CTCs), circulating tumor DNA (ctDNA), RNA, extracellular vehicles (EVs), and proteins, provide information about potential metastatic lesions, and reveal the cancer profile [6,7].

CTCs play a key role in the spread of metastases as they enter the bloodstream and migrate to form distant lesions [8]. CTCs that migrate into the bloodstream exhibit both epithelial and mesenchymal properties [9]. The mesenchymal nature of CTCs can be an obstacle to their isolation [9,10], as many methods are epithelial-dependent, such as CellSearch^TM^, the gold standard for detection and isolation of CTCs, particularly in breast [11], colorectal [12], and prostate cancer [13]. CTCs can survive either as individual cells or as cell clusters that can remain inactive for many years until they develop recurrences or metastasis, indicating the intolerance of CTCs to the lethal effects of existing therapeutic approaches [14,15].

Programmed cell death protein 1 (PD-1) is normally located on the surface of activated T cells, and the corresponding ligand (PD-L1) is physiologically expressed on macrophages, dendritic cells, natural killer cells, B lymphocyte cells, and vascular endothelial cells [16]. In patients, PD-L1 is expressed in cancer cells and inhibits the cytotoxic effect of CD8+ T cells against them [3,16]. Several studies have shown increased PD-L1 protein expression in CTCs of different cancer types, including prostate cancer, which is potentially related to patient outcomes [15,16,17,18,19]. Furthermore, the evolution of molecular assays for the detection of *PD-L1* mRNA expression has shown that the detection of *PD-L1* expression in CTCs is practicable and can provide real-time clinical information [20]. In head and neck squamous cell carcinoma (HNSCC), the detection of CTCs overexpressing *PD-L1* using an RT-qPCR method at the end of definitive non-surgical treatment including chemoradiation correlates with a lower probability of achieving complete response (CR) and a higher risk of relapse and death [21].

CXCR4 is normally involved in homeostatic processes in the bloodstream, such as leukocyte transport and hematopoiesis [22]. On the other hand, it may also be involved in carcinogenesis and metastasis in various cancers, including prostate cancer [22,23,24]. This is confirmed by a meta-analysis showing that increased CXCR4 expression in prostate cancer samples correlates to metastases [25].

JUNB is a transcription factor that can act as an oncogene and induce abnormal proliferation of quiescent cells [26]. In the prostate cancer cell line PC-3, JUNB has been indicated to promote migration and antagonize the opposite function of JUND [27]. In addition, a full transcriptome analysis showed that JUNB is present in severely affected patient groups and metastatic lesions, which are thought to play a role in prostate cancer progression [2]. On the other hand, JUNB demonstrates a new upstream signaling role triggering the activation of p16/pRb, which is crucial for the initiation and maintenance of senescence. Additionally, activated JUNB effectively hinders the malignant progression of prostate cancer as measured by invasion and metastasis [28].

Both CXCR4 and JUNB are highly expressed in CTCs isolated from metastatic breast [29] and non-small cell lung cancer (NSCLC) patients [30], as well as in disseminated tumor cells (DTCs) isolated from early-stage breast cancer migrated in bone marrow [31]. The presence of JUNB in CTCs from breast cancer patients is correlated to poor prognosis [29]. In addition, CXCR4 and JUNB have been associated with poorer overall survival (OS) of early-stage breast cancer patients [31] and lower progression-free (PFS) and overall survival of NSCLC patients [30].

In the current study, we investigated for the first time the expression, at the RNA and protein level, of CXCR4, JUNB, and PD-L1 in CTCs from (mPCa) patients. Furthermore, we investigated the correlation between the results on protein and RNA levels and the clinical relevance of the different CTC phenotypes and mRNA patterns with the clinical outcome.

## 2. Materials and Methods

### 2.1. Characteristics of the Patients

Patients’ characteristics for every cohort, depending on the analyzed method used, are presented in Appendix A. A total of 48 patients with mPCa and 10 healthy donors were included in the study. The 48 patients provided blood collection for CTC identification through Ficoll density gradient isolation (Appendix A). A total of 25 out of 48 patients were additionally evaluated for mRNA analysis (Appendix A), and 17 out of 48 patients were additionally evaluated for CTC identification through ISET system isolation (Appendix A). The mean age of the 48 patients was 75 years (range: 42–87 years), of the 25 patients was 73 years (range: 44–87 years), and of the 17 patients was 74 years (range: 65–83 years). A total of 38 patients were treatment naïve (baseline sample), and 10 patients were enrolled before the initiation of second-line treatment. The protocol was approved by the ethics and scientific committees of all participating institutions [Venizeleion, General Hospital, (136/26/26-10-2022); Metropolitan General Hospital, (172/18-09/2020); Larissa General University Hospital, (32710/3-8-20); ST Andrews General Hospital of Patras, (521/13-10-2020)]. The patients and the healthy donors provided their written informed consent for blood collection and for the use of clinical outcome information for research endeavors.

### 2.2. Cell Culture

H1299, PC-3, and MDA-MB-231 cell lines were used as control cells to study CXCR4, JUNB, and PD-L1 expression and obtained from the American Type Culture Collection (Manassas, VA, USA). H1299, PC-3, and MDA-MB-231 cells were cultured in Dulbecco’s Modified Eagle Medium with Glutamax (Thermo Fisher Scientific, Waltham, MA, USA) supplemented with 10% fetal bovine serum (FBS; PANBiotech, Aidenbach, Germany) and 50 U/mL penicillin/50 g/mL streptomycin (Thermo Fisher Scientific, Waltham, MA, USA). Cells were maintained at 37 °C in a humidified atmosphere with 5% CO_2_ in air and subcultured with 0.25% trypsin-EDTA (Thermo Fisher Scientific, Waltham, MA, USA).

### 2.3. CTC Analysis

#### 2.3.1. Protein Analysis

##### Blood Sampling and Cytospin Preparation in Patients

Ten mL of peripheral blood was collected from all patients and from 10 healthy donors, using K2EDTA tubes. All blood samples were collected by mid-vein puncture, and the first 5 mL was discarded to avoid contamination with skin epithelial cells. Peripheral blood mononuclear cells (PBMCs) from mPCa patients were isolated by Ficoll-Hypaque (d = 1.077 g/mol) density centrifugation at 1800 rpm for 30 min without dissertation. The Ficoll density method enables isolation of CTCs contained in the mononuclear fraction of peripheral blood mononuclear cells, which is size- and EpCAM-independent. After washing twice with PBS and centrifuging at 1500 rpm for 10 min, aliquots of 500,000 patients’ PBMCs/500 μL were cyto-centrifuged at 2000 rpm for 2 min on Superfrost glass slides (Thermo Fisher Scientific, Waltham, MA, USA). Cytospins were dried and stored at −80 °C.

##### ISET Isolation

In a subgroup of 17 mPCa patients, who were able to provide additional 10 mL of blood, ISET (Isolation by Size of Epithelial Tumor cells) technology was applied. This device employs size-based, label-independent isolation to effectively capture CTCs with diverse phenotypic profiles, as CTCs usually have a larger size compared to most leukocytes [32]. Following the manufacturer’s guidelines, CTC isolation was performed within 2 days from blood sampling, using 10 mL of peripheral blood collected in K2EDTA tubes. The blood was diluted in a 1:10 ratio with ISET buffer (Rarecells, Paris, France) for 10 min at room temperature, facilitating erythrolysis and maintaining CTC integrity. The diluted samples were then filtered through the ISET membrane at a depression of 10 kPa. Fixed cells, including CTCs unable to pass through the pores (8 μm), were held on the membrane, creating 10 spots. The threshold of method’s detection, specified by the manufacturer, is 1 CTC per 10 mL of blood. For each patient, a spot containing 10^6^ cells was utilized for CTC identification and the assessment of PD-L1 expression. This was accomplished by employing triple immunofluorescence staining, subsequently followed by VyCAP microscopy analysis.

##### Triple Immunofluorescence Analysis

One slide from each patient was used for triple immunofluorescence staining to identify CTCs and assess PD-L1 expression, using antibodies against cytokeratins (CK), PD-L1, and CD45 (CK/PD-L1/CD45). The presence of CK, using an A45-B/B3 antibody for the detection of CK-8, CK-18, and CK-19, was used to characterize a nucleated cell as CTC. CD45 expression was used as a negative hematopoietic biomarker expressed only in PBMCs. Many CTCs with a more mesenchymal nature may have decreased levels of CK, so, additionally, the cytomorphologic criteria described by Meng et al. (high nuclear/cytoplasmic ratio, larger cells than white blood cells, irregular nuclear shape, and size) were applied to identify a cell as a CTC [32]. Furthermore, morphological characteristics of CTCs described by Adam et al. and Park et al. were observed in subpopulations of CTCs detected in the present study [33,34]. According to the CTC morphology observed on the (CK/PD-L1/CD45) staining, a second slide from each patient was analyzed for the assessment of CXCR4 and JUNB expression in CTCs, using antibodies against CK, CXCR4, and JUNB (CK/CXCR4/JUNB), as we have previously reported [19,30]. 

Positive and negative controls were prepared for each experiment using H1299 or PC-3 cells spiked into PBMCs from healthy volunteers (1000 H1299 cells or PC-3 cells/100,000 PBMCs) to imitate CTCs’ microenvironment in patients’ cytospins slides. Negative controls were generated by excluding the incubation of corresponding primary antibodies and incubating the cells with the corresponding secondary antibody. Each experiment included three negative controls (one negative for every primary antibody used) and one positive control for all antibodies, to assess methods’ sensitivity and specificity.

Slides from both triple immunofluorescence stainings were analyzed using the VyCAP system (VyCAP B.V., Enschede, The Netherlands). The VyCAP system operates as an imaging platform, where cytospins slides can be scanned automatically using four different channels. In the current study, we used DAPI, CK, PD-L1, CD45 for the first immunofluorescent staining and DAPI, CK, JUNB, CXCR4 for the second immunofluorescent staining. The corresponding frames were analyzed to identify and characterize patients’ CTCs. Additionally, the corresponding frames of patients’ slides were also double-checked using the ACCEPT software (automatic software for CTCs detection, University of Twente, Enschede, The Netherlands). Images for control samples and CTCs were captured based on the determined exposure time and examined on negative and positive controls for each antigen. The identification of CTCs was conducted blind to clinical data.

Results were presented as percentage of patients expressing specific phenotype among the total CTC-positive patients. Specifically, percentages of patients for a specific phenotype were calculated as follows:Percentage of patients = (number of patients with the phenotype/total CK-positive patients) ∗ 100%,(1)

Regarding the percentage of CTCs to address the variability in the number of isolated CTCs per patient and ensure that each patient’s contribution was weighted equally, we evaluated the percentages of CTCs corresponding to specific phenotypes for each patient individually. This involved calculating the percentage of CTCs exhibiting a specific phenotype within the total CTCs of each patient. Here is how this was performed:Frequency of CTCs per patient = (patient CTCs with the phenotype/total patient CTCs) ∗ 100%,(2)

Then the average of all patients’ frequencies was calculated, for every phenotype.

#### 2.3.2. mRNA Analysis

##### CTC Isolation, RNA Extraction, and cDNA Synthesis

In total, 25 mPCa patients and 10 healthy individuals were included, and 10 mL of peripheral blood was collected from all of them for mRNA analysis. Prior to analysis, the PB was immediately mixed by gentle inversion 10 times. Following the addition of 30 mL of red cell lysis buffer (containing NH_4_Cl, 155 mmol/L; KHCO_3_, 10 mmol/L; and EDTA, 0.1 mmol/L, pH = 7.3), samples were incubated for 20 min at room temperature, with occasional gentle inversion for mixing. After centrifugation of 530× *g* at room temperature for 20 min, the supernatant was removed, and, subsequently, red cell lysis buffer (30 mL) was added. After centrifugation (530× *g*, room temperature, 10 min), the supernatant was removed, and 10 mL of red cell lysis buffer was added. Magnetic beads, coated with the monoclonal antibody BerEP4 against the human epithelial antigen EpCAM, were further used for CTC enrichment (Dynabeads® Epithelial Enrich, Life Technologies, Waltham, MA, USA) [35].

CTCs’ total RNA was extracted using TRIZOL-LS (ThermoFisher Scientific, Waltham, MA, USA), followed by cDNA synthesis according to previously established protocols [35,36].

##### External Quality Controls

For the development of molecular assays for *CXCR4* and *JUNB*, synthetic standards corresponding to two different concentrations were used: one low (10 copies) and one medium (10^3^ copies). By using these two different positive controls, we selected the optimal concentration of each component that ensured the best performance of the two assays. The analytical linearity and sensitivity of the two RT-qPCR assays for *JUNB* and *CXCR4* were evaluated by analyzing two different types of external standards: (a) synthetic DNA standards ranging from 10^5^ copies/μL to 10 copies/μL (Appendix A) and (b) cancer cell lines ranging from 10^4^ cells to 10 cells (Appendix A). H1299 was served as a positive control for mRNA expression of JUNB, while MDA-MB-231 was utilized as a positive control for mRNA expression of CXCR4.

##### RT-qPCR

The detection of mRNA of *PD-L1*, *CK-19 46*, *CK-8*, *CK-18*, and the reference gene of beta-2-microglobulin (*Β2Μ*) [37], which was conducted through RT-qPCR assays to confirm the quality of the analyzed samples, has been reported previously. Two novel RT-qPCR assays were developed for the detection of *CXCR4* and *JUNB* mRNA transcripts and validated for analytical specificity and sensitivity. The development of two single RT-q PCR assays was relied on an in silico design of highly specific probes and primers. The in silico design was performed using Primer Premier 5.0 software (Premier Biosoft, San Francisco, CA, USA). Homology searches in the nucleotide database (NCBI, Nucleotide BLAST) were used as an initial evaluation of specificity of the primers and probes. We also designed two synthetic DNA oligos that we used as standards for the analytical validation of RT-qPCR assays for the detection of *CXCR4* and *JUNB* mRNA transcripts.

The amplification reaction mix for *CXCR4* and *JUNB* included 2 μL PCR synthesis buffer (5Χ), 1.0 μL MgCl_2_ (25 mM), 0.15 μL dNTPs (10 mM), 0.1 μL hot start DNA polymerase (HotStart, 5 U/μL, Promega, Fitchburg, WI, USA), 0.3 μL forward and reverse primers (10 μΜ), 1.0 μL hydrolysis probe (3 μM). For the CXCR4 amplification reaction mix, 2 μL PCR synthesis buffer (5Χ), 1.0 μL MgCl_2_ (25 mM), 0.15 μL dNTPs (10 mM), 0.1 μL Hot Start DNA Polymerase (HotStart, 5 U/μL, Promega, Fitchburg, WI, USA), 0.3 μL forward and reverse primer (10 μΜ), 0.83 μL hydrolysis probe (3 μM) were added. To each PCR mix, 1 μL cDNA was added, followed by the addition of dH_2_O to achieve a final volume of 9 μL. The two protocols had common conditions: 1 cycle at 95 °C for 2 min, followed by 45 cycles of 95 °C for 10 s, annealing at 58 °C for 40 s, and a final cooling cycle at 40 °C for 30 s. All RT-qPCR reactions were achieved in the LightCycler® z480 (Roche Diagnostics, Mannheim, Germany) according to MIQE guidelines [38]. The expression levels of *CXCR4*, *JUNB*, and *PD-L1* were normalized with respect to the expression of *Β2Μ* [21,39].

##### Normalization for mRNA Expression Data

Normalization to quantify the expression of *CXCR4*, *JUNB*, and *PD-L1* in CTCs was performed using the 2^−ΔΔCq^ method for the expression of *B2M* as a reference gene [39]. More specifically, the expression of the *CXCR4*, *JUNB*, and *PD-L1* genes was estimated as a relative ratio to the expression of *B2M* both in the CTC fraction of the patients and in the corresponding CTC fraction of the healthy donor group used as a calibrator.

### 2.4. Statistical Analysis

The statistical software SPSS version 27 (IBM, Armonk, NY, USA) was used for statistical tests at the significance level *p* < 0.05. Progression-free survival (PFS) was estimated as the time within patient enrollment in the study and disease progression. Overall survival (OS) was estimated as the time within patient enrollment and last follow-up or death from any cause. In order to determine the impact of each CTC phenotype on PFS, univariate Cox regression analysis was performed. Kaplan—Meier analysis was used to investigate the presence of CTC numbers or specific CTC phenotypes in correlation with patients’ clinical outcomes. To compare Kaplan—Meier curves and Cox regression analysis for PFS and OS, the log-rank test was used. χ^2^ was performed to investigate whether there was a statistically significant difference between the observed phenotypes with PD-L1 in Ficoll density gradient isolation versus ISET isolation. The Wilcoxon rank test was performed to examine quantitative differences between the median values of all tested groups. The Mann—Whitney test was performed to determine the differences between the median fold changes of *CXCR4*, *JUNB*, and *PD-L1* levels between groups.

## 3. Results

### 3.1. Triple Immunofluorescence Analysis

#### 3.1.1. PD-L1 Protein Expression in CTCs from mPCa Patients

Representative images of a CTC isolated from an mPCa patient with positive staining for PD-L1 is shown in Figure 1. Additionally, representative images of positive controls with spiked H1299 (1000 cells) or PC-3 (1000) cells in 100,000 PBMCs from healthy donors are presented in Appendix A. CK-positive cells could not be detected in the samples of normal donor blood without the presence of spiked cancer cells.

Forty-eight patients with mPCa were analyzed. The average CTC count for this immunofluorescent staining was 0.67, and the range was 0–6. CTCs were detected in 37.5% of patients (18 of 48). Among the CK-positive patients, 61.1% (11 of 18) had the (CK+/PD-L1+/CD45-) phenotype and 38.9% (7 of 18) had the (CK+/PD-L1-/CD45-) phenotype (Figure 2a). Regarding the average percentage of total identified CTCs per patient, 61.1% belonged to the (CK+/PD-L1+/CD45-) phenotype, and 38.9% were characterized as (CK+/PD-L1-/CD45-) (Figure 2b). It is noteworthy that the patients did not harbor simultaneously PD-L1-positive and PD-L1-negative CTCs, as is shown in Appendix A. Statistical analysis revealed that there were no significant differences between the phenotypes (CK+/PD-L1+/CD45-) and (CK+/PDL1-/CD45-) (p = 0.343) (Appendix A).

Out of the 48 patients, 17 were investigated for the detection of CTCs using the ISET platform, since we have previously shown that ISET has an impressively higher recovery rate compared to the Ficoll density gradient [40]. The average CTC count for this immunofluorescent staining at the ISET spot per patient was 7 and the range was 0–97. Larger number compared to samples processed through Ficoll density isolation. CTCs were detected in 47.1% of the patients (8 out of 17). Among the CK-positive individuals, 75.0% (6 out of 8) exhibited the (CK+/PD-L1+/CD45-) phenotype, while 62.5% (5 out of 8) displayed the (CK+/PD-L1-/CD45-) phenotype (Figure 2c). Notably, 37.5% (3 out of 8) of the CK-positive patients manifested both observed phenotypes, indicating the coexistence of CTCs with and without PD-L1 expression in the same individual, arising from ISET enrichment (Appendix A). In terms of the average frequency of the total isolated CTCs per patient, 55.1% were characterized as (CK+/PD-L1+/CD45-) and 44.9% as (CK+/PD-L1-/CD45-) (Figure 2d). There was no significant difference between the frequencies of the two distinct phenotypes using the ISET system, (CK+/PD-L1+/CD45-) and (CK+/PD-L1-/CD45-): *p* = 0.861 (Appendix A). No statistically significant difference was shown between the Ficoll and ISET methods for the number of total CTCs and the identified phenotypes of (CK+/PD-L1+/CD45-) and (CK+/PD-L1-/CD45-) through the Wilcoxon rank test (*p* = 0.591, *p* = 0.714, and *p* = 0.206, respectively) and χ^2^ (*p* = 0.057, *p* = 0.644, and *p* = 1, respectively).

No statistically significant differences were observed between treatment-naive patients and patients who had yet to undergo initiation of second-line treatment regarding PD-L1 expression.

#### 3.1.2. CXCR4 and JUNB Protein Expression in CTCs from mPCa Patients

Representative images of a CTC isolated from an mPCa patient with simultaneous expression of CXCR4 and JUNB is shown in Figure 3. In addition, positive controls with spiked H1299 or PC-3 cells in 100,000 PBMCs from healthy donors are presented in Appendix A.

Forty-eight patients with mPCa were analyzed. The average CTC count for this immunofluorescent staining was 0.9, and the range was 0–16. CTCs were detected in 33.3% of patients (16 of 48). Of the CTC-positive patients, 62.5% (10 of 16) had the (CK+/CXCR4+/JUNB+) phenotype, 18.8% (3 of 16) had the (CK+/CXCR4-/JUNB+) phenotype, and 25.0% (4 of 16) had the (CK+/CXCR4+/JUNB-) and the (CK+/CXCR4-/JUNB-) phenotype (Figure 4a). Regarding the average percentage relative to the total CTC count per patient, 52.5% belonged to the (CK+/CXCR4+/JUNB+) phenotype, while the percentages of the rest of the phenotypes were 13.7% for (CK+/CXCR4-/JUNB+), 17.0% for (CK+/CXCR4+/JUNB-), and 16.8% for (CK+/CXCR4-/JUNB-) (Figure 4b). Most of the patients (81.3%) harbored only one phenotype, while more than one phenotype was observed in three patients (18.8%) (Appendix A). Referring to CXCR4 and JUNB expression alone, 75.0% of the patients were positive for CXCR4 and 75% were also positive for JUNB (Figure 4c). For average percentages of the total CTCs observed in the mPCa patients, 74.2% were CXCR4-positive, while 70.6% were JUNB-positive (Figure 4d). For the phenotypes of (CK/CXCR4/JUNB) staining, there was a significant quantitative difference with the Wilcoxon rank test between the phenotypes of (CK+/CXCR4+/JUNB+) and (CK+/CXCR4-/JUNB+) (*p* = 0.013) (Appendix A).

No statistically significant differences were observed between treatment-naive patients and patients who had yet to undergo the initiation of second-line treatment regarding CXCR4 or JUNB expression.

### 3.2. Molecular Analysis

#### 3.2.1. Limit of Detection (LOD) and Limit of Quantification (LOQ) of RT-qPCR for *CXCR4* and *JUNB*

The LOD and LOQ were determined using absolute quantification of low-input calibrators. In all cases, the LOD was shown to correspond to 3 copies/μL. The limit of quantification (LOQ), defined as three times the LOD, corresponds to 9 copies/μL. This is consistent with the MIQE guidelines for RT-qPCR assay development [38], according to which the minimum number of copies in a sample that can be accurately measured is equivalent to 3 copies/μL [36]. In addition, the assays developed demonstrated linearity across the entire quantification range (10^5^-10 copies) and correlation coefficients exceeding 0.99 in all cases, which indicated a precise log-linear relationship.

#### 3.2.2. Relative Fold Change of *CXCR4*, *JUNB* and *PD-L1* Levels

Before proceeding with the combined gene expression analysis, the quality of all cDNAs of the two different groups of samples was checked by RT-qPCR for *B2M* (reference gene) (Appendix A). *B2M* expression is used as an internal control for sample quality to avoid false negative results but also as a reference gene for relative quantification. As shown in Appendix A, the median *B2M* Cq value in the HD group was 22.52, while it was 22.45 in the cancer group (Mann—Whitney test, Ζ = −0.530, *p* = 0.596). The Kruskal—Wallis test was the same between the two different sample groups (Kolmogorov—Smirnov, Z = 0.535, *p* = 0.938) (Appendix A). 

The median fold change in *CXCR4*, *JUNB*, and *PD-L1* expression in the respective positive groups was 3.36 (range: 2.77–6.92), 8.17 (range: 3.81–26.35), and 8.34 (range: 4.29–33.82), respectively, while the median fold change in the negative groups was as follows: *CXCR4*: 1.62 (range: 0.78–2.62) (Mann—Whitney test, Ζ = −4.215, *p* < 0.001); *JUNB*: 0.73 (range: 0.12–2.83) (Mann—Whitney test, Ζ = −3.813, *p* < 0.001); *PD-L1*: 1.59 (range: 0.0–3.34) (Mann—Whitney test, Ζ = −4.243, *p* < 0.001) (Figure 5). There was a significant difference observed in the median fold change of *CXCR4*, *JUNB*, and *PD-L1* expression between the samples from HD and the positive samples (Mann—Whitney test, *CXCR4*: Ζ = −3.873, *p* < 0.001, *JUNB*: Ζ = −3.416, *p* < 0.001, *PD-L1*: Ζ = −3.838, *p* < 0.001). The median fold change of *CXCR4*, *JUNB*, and *PD-L1* expression between the HD and negative samples was not significantly different (Mann—Whitney test, *CXCR4*: Ζ = −1.376, *p* = 0.169, *JUNB*: Ζ = −0.791, *p* = 0.429, *PD-L1*: Ζ = −1.036, *p* = 0.300).

#### 3.2.3. Molecular CTC Profile of mPCa Patients

In 25 of these patients, we had material available for subsequent molecular analysis after CTC isolation. The first 5 mL of blood was discarded for CTC isolation from PB to prevent contamination of skin epithelial cells. The peripheral blood was then collected in a 10 mL K2EDTA tube (BD Vacutainer, Plymouth, UK) at the blood collection centers and transported to the ACTC laboratory. CTC isolation was performed within 2 days, as CTCs are detectable as early as 48 h after blood collection when the PB is stored in K2EDTA tubes.

In the cohort of the examined mPCa patients, expression of *CK-19* was detected in 4/25 (16.0%) samples, expression of *CK-8* in 19/25 (76.0%) samples, and expression of *CK-18* in 4/25 (16.0%) samples. At least one cytokeratin mRNA was expressed in 19/25 (76.0%) samples. As for the remaining genes, 11/25 (44.0%) were positive for overexpression of *CXCR4*, 7/25 (28.0%) were positive for overexpression of *JUNB*, and 12/25 (48.0%) were positive for overexpression of *PD-L1* (Figure 5), compared to the healthy male donor group. A total of 9/11 (81.8%) samples overexpressing *CXCR4* were also positive for CK, 4/7 (57.1%) samples overexpressing *JUNB* were also positive for CK, and 8/12 (66.7%) samples overexpressing *PD-L1* were also positive for CK. In the majority (14/19, 73.7%) of CK-positive samples, at least one of *CXCR4*, *JUNB*, or *PD-L1* was also expressed. The pooled results of the expression of all the genes tested are shown in Appendix A, while the relative expression of the genes is shown in Appendix A.

We used the Wilcoxon signed-rank test for related samples to determine quantitative variations between the median values of the relative fold changes for each gene. The median differences between the analysis of mRNA expression of *JUNB* and *CXCR4* (*p* = 0.979) and the analysis of mRNA expression of *JUNB* and *PD-L1* (*p* = 0.06) were not found to be significant. However, when analyzing the relative fold-change data between *PD-L1* and *CXCR4* (*p* = 0.020), significant differences were found (Figure 6).

### 3.3. Combined Analysis between Molecular and Protein Analysis

Our results from the molecular CTC analysis were compared with the immunofluorescence analysis. The agreement (negative and positive samples) between mRNA and protein expression was (a) for JUNB, 17/25 samples (68.0%), (b) for CXCR4 expression, 12/25 (48.0%), (c) for PD-L1 expression, 13/25 (52.0%). Our results showed a higher sensitivity for the combination of molecular and IF analysis. Specifically, the positivity rate in these 25 patients increased for JUNB to 10/25 (40.0.0%), the positivity rate for CXCR4-positive patients increased to 15/25 (60.0%), and the positivity rate for PD-L1-positive patients increased to 17/25 (68.0%) (Appendix A).

### 3.4. Clinical Significance

Survival analysis was performed for all observed molecular and protein profiles. The univariate Cox regression analysis for the Ficoll density gradient isolation, for 23 patients out of 48 with available data of PFS duration, showed a significantly higher risk of relapse in the group of patients harboring CTCs of the (CK+/CXCR4+/JUNB-) phenotype (quantitative variable) (HR: 2.5, *p* = 0.049). No statistically significant correlation was found in the survival analysis for PD-L1 in the Ficoll density gradient isolation.

Regarding the ISET isolation, 13 patients had available data on PFS duration, and 15 patients out of 17 had available data on OS duration. Kaplan—Meier analysis indicated a significantly shorter PFS (HR: 1137.5, *p* = 0.046) for 2 patients with ≥2 CTCs compared to 11 patients who had ≤1 CTCs detected (3 months with a range of 1.6–4.4 vs. 7.5 months with a range of 5.8–9.1), and a decreased OS (HR: 9.0, *p* = 0.031) for 3 patients with ≥2 CTCs compared to 12 patients who had ≤1 CTCs detected (3.7 months with a range of 0.6–6.7 vs. 12.8 months with a range of 10.5–10) (Figure 7a,b). Furthermore, the detection of ≥1 CTC in six patients was also associated with lower OS compared to nine patients who had no CTCs (HR: 115.6, *p* = 0.032) (3.5 months with a range of 1–14 vs. 5 months with a range of 0–10) (Figure 7c).

In addition, a significantly lower OS through Kaplan—Meier analysis in ISET isolation was observed for 4 patients harboring CTCs positive for PD-L1 (CK+/PD-L1+/CD45-) contrary to 11 patients without this phenotype (HR: 262.7, *p* = 0.007) (3 months with a range of 1–14 vs. 4 months with a range of 0–10) (Figure 7d). No other correlations were found between CTC numbers and phenotypes with other patients’ characteristics.

## 4. Discussion

The diversity of CTCs regarding their morphological features, protein expression patterns, and molecular profile has been previously examined by many studies [33,34,41,42]. To date, there are many different methods for CTC isolation based on distinct biological or morphological characteristics [40,43,44], each of which isolates subpopulations and not all the existing CTCs. This fact makes the detection of CTCs a major challenge [9,10]. Furthermore, previous studies by our group have indicated a high expression of CXCR4, JUNB, and PD-L1 in CTCs and/or DTCs of breast cancer and NSCLC patients, related to their survival [29,30,31]. In the present study, we combined the results of different CTC isolation methods, (a) an EpCAM-independent method (Ficoll) [19,30] and (b) an EpCAM-dependent method [35,36], in association with two different detection methods, (a) immunofluorescence analysis [19,30] and (b) molecular analysis [21], for a more holistic approach regarding the expression status of the above biomarkers in the CTCs of prostate cancer patients.

In 17 patients with enough volume of blood for further analysis, we also performed CTC isolation through the size-based ISET technology, which raised the CTCs’ detection rate from 37.5% (18 out of 48) to 47.1% (8 out of 17), in line with our previous study [40]. A higher recovery rate through ISET filtration compared to Ficoll was also obvious for the absolute number of CTCs [7 CTCs (range 0–97) vs. 1 CTC (range 0–6)]. However, no statistically significant differences were found between the Ficoll and ISET methods for the number of total CTCs and the (CK+/PD-L1+/CD45-) and (CK+/PDL1–/CD45-) phenotypes, through the Wilcoxon rank test (*p* = 0.591, *p* = 0.714, and *p* = 0.206, respectively) and χ^2^ test (*p* = 0.057, *p* = 0.644, and *p* = 1, respectively). This may be owed to the smaller number of patients enrolled in ISET isolation compared to the number of patients enrolled in Ficoll density isolation (17 versus 48). Further studies with larger patient cohorts will be needed to confirm these results. In samples processed through the Ficoll density gradient, simultaneous identification of PD-L1-positive and PD-L1-negative CTCs was not observed. These results could be attributed to the limited number of CTCs isolated with the Ficoll density gradient [40]. Conversely, the ISET system evades this issue by isolating larger numbers of tumor cells. The ISET platform facilitates the CTCs’ enrichment, as the membrane mostly holds cells with diameters equal to or exceeding 8 μm, while other blood cells and constituents are discarded. Employment of the ISET isolation enabled the observation of PD-L1 heterogeneity in CTCs of prostate cancer patients. Particularly, 37.5 of the CTC-positive patients harbored both phenotypes (PD-L1-positive and PD-L1-negative) in their blood, providing more informative results regarding tumor cells metastasis.

CellSearch^TM^ stands as the only FDA-approved system for identifying CTCs retrieved from individuals diagnosed with mPCa [45,46,47,48]. However, the detection of CTCs with a mainly mesenchymal phenotype, due to epithelial—mesenchymal transition (EMT), characterized by the absence of EpCAM, may be inadequate with CellSearch^TM^ [10,33,42]. An earlier study involving 34 human breast cancer cell lines revealed the inability of CellSearch^TM^ to identify normal-like breast cancer cell lines [49]. Moreover, in another study, a greater number of CTCs was identified in castration-resistant patients, using an antibody-targeting cell-surface vimentin, in contrast to CellSearch^TM^ [10]. It has been reported that different detection systems could complement the EpCAM-based system of CellSearch^TM^ by improving the overall rate of CTC detection [33,50,51]. For instance, a recent report involving 30 patients with breast and prostate cancer evaluated CTC isolation using a microfiltration system (CellSieve) and the CellSearch^TM^ system. The study revealed that the CellSieve yielded a higher number of isolated CTCs with different morphological features, while the CellSearch^TM^ system detected a subpopulation of CellSieve CTCs [33]. Furthermore, the presence of CTCs detected using CellSearch^TM^ as an EpCAM-dependent method and two EpCAM-independent detection systems is predictive of decreased OS in patients with metastatic breast cancer [52]. Our study is in line with the above notions, as the positivity rate combining mRNA analysis and immunofluorescent stainings for (CK+JUNB+), (CK+CXCR4+), and (CK+PD-L1+) cells was increased (Appendix A).

Our results indicated that the (CK+/PD-L1+/CD45-) phenotype is predominant in patients with mPCa using both Ficoll density gradient method (61.1%) and ISET filtration (75.0%). Interestingly, in ISET isolation the (CK+/PD-L1+/CD45-) phenotype was correlated with decreased OS with a Kaplan—Meier analysis (HR: 262.7, *p* = 0.007). These results are in line with a previous study on triple-negative breast cancer (TNBC) patient CTCs showing that the expression of PD-L1 is linked to lower OS in Kaplan-Meier analysis (HR: 8.7, *p* < 0.001) [19]. Furthermore, the investigation of PD-L1 expression in CTCs using the CellSearch^TM^ platform has shown that PD-L1 is frequently expressed in metastatic breast cancer CTCs (68.8%) [18]. Similar results have been shown for advanced NSCLC patients, indicating that PD-L1-positive CTCs are correlated to poor prognosis [53]. In prostate cancer, Zhang et al. have shown that at least one PD-L1-positive CTC was detected at baseline in 40% of men with metastatic hormone-sensitive prostate cancer (mHSPC), 60% of men with metastatic castration-resistant prostate cancer (mCRPC) starting abiraterone acetate/prednisone or enzalutamide (pre-ARSI), and 70% of men with mCRPC post-ARSI [17]. In addition, a study investigating the expression of PD-L1 on circulating epithelial tumor cells (CETCs) of different types of cancer, including prostate cancer, showed that PD-L1 is expressed in 100% of CETCs from prostate cancer patients [16]. The detection of PD-L1 in prostate cancer could lead to the customization of immunotherapies [54], and its combination with other drugs could bring encouraging results and benefits [3]. Therefore, our study provides an interesting protocol for the evaluation of PD-L1 in CTCs from PC patients with potential clinical significance.

Furthermore, our data showed that both CXCR4 and JUNB were expressed at the protein level in the CTCs of most CK-positive mPCa patients. Similar observations regarding high expression of CXCR4 and JUNB were noted in our prior investigations involving metastatic [29] and early breast cancer patients [31], as well as in NSCLC patients [30]. The most frequent phenotype was (CK+/CXCR4+/JUNB+) (62.5%), as we have previously shown (90% in breast cancer and 50% in NSCLC) [30,31], and (CK+/CXCR4-/JUNB+) showed the lower percentage of positivity (18.8%) in contrast to the breast cancer [31] and NSCLC study [30], where (CK+/CXCR4+/JUNB-) was the rarest phenotype (5% and 6%, respectively). The number of CTCs expressing the (CK+/CXCR4+/JUNB-) phenotype was associated with poorer PFS in the Cox regression analysis (HR: 2.5, *p* = 0.049). Consistent with this observation, in extensive-stage small cell lung cancer (SCLC) patients, the presence of ≥4 CXCR4-positive CTCs was correlated to decreased OS (HR: 5.01, *p* = 0.041) [30]. Recently, CXCR4-positive CTCs were shown to persist in the blood of prostate cancer patients treated with radiotherapy for up to 3 months, suggesting that CXCR4-positive CTCs may represent aggressive CTC subclones that contribute to treatment resistance [23]. CXCR4 antagonists could be used in a stratified manner in the treatment of CRPC [55,56]. In PC-3 tumor xenograft models, AMD3100 has been shown to inhibit tumor growth and reduce microvessel formation by inhibiting CXCR4/Akt signal transduction [55], and CTCE-9908 has been shown to reduce tumor spread and angiogenesis in an orthotopic prostate cancer model [56]. Therefore, anti-CXCR4 treatment in prostate cancer could be a potential alternative, and our protocol may provide a companion diagnostic test.

Gene expression analysis for *PD-L1*, *CXCR4*, and *JUNB* was performed on a small cohort of 25 patients. In an earlier study by Zavridou et al., *PD-L1* mRNA expression in EpCAM-positive CTCs and plasma-derived exosomes isolated from mCRPC patients was investigated in a comparative study [35]. In addition, the expression of AR splice variants of mCRPC in CTCs and exosomes was also investigated [36]. Many studies have shown that sequential biopsies provide important prognostic and predictive data based on *PD-L1* status changes [57,58,59]. We have already developed and clinically tested a molecular assay to detect *PD-L1* transcripts in CTCs from HNSCC patients, which offers prognostic information [21]. According to our results, almost half of the patients with mPCa were found to overexpress *PD-L1*. High PD-L1 expression of over 50% was also found in NSCLC CTCs during radiotherapy [60].

In addition, we have developed two novel RT-qPCR assays for the detection of mRNA expression of *CXCR4* and *JUNB*, which are characterized by their analytical sensitivity and specificity. Molecular assays have the advantage of detecting gene expression at very low levels, as we have previously shown [36]. A high prevalence of *CXCR4* transcripts compared to *JUNB* transcripts was observed. This is the first time that a molecular assay has been developed and clinically evaluated for the detection of *CXCR4* and *JUNB* transcripts in mPCa CTCs.

We compared the median differences between the analysis of mRNA expression of all the genes tested. We found no significance between the relative fold-change data of *JUNB* and *CXCR4* and *JUNB* and *PD-L1*. Previous work in metastatic breast cancer has shown that CXCR4 and JUNB are linked in a common signaling pathway by bioinformatic analysis [29]. Furthermore, JUNB can dock in the region of the PD-L1 promoter and thus enable transcription of the *PD-L1* gene [61]. In addition, when analyzing the relative fold-change data between *PD-L1* and *CXCR4*, significant differences were found. The CXCR4—CXCL12 axis regulates the transmission of diverse downstream signaling pathways crucial for tumor cell survival, proliferation, and migration [62], and PD-1/PD-L1 controls immune tolerance [63]. Recent data have shown that a combination approach of drugs targeting the function of CXCR4 and PD-L1 enhances the efficacy of anti-PD-L1 therapy in TNBC by preventing two important signaling pathways of tumor proliferation [64].

The very low frequency of CTCs among blood cells makes the Isolation and detection methods particularly challenging [40]. We also performed a combined investigation in molecular and protein levels for the three analyzed common gene targets. Results indicate an increased detection rate of cytokeratin-positive patients through RT-qPCR compared to IF for the cohort of 25 patients (76% vs. 60%), probably attributed to the fact of an increased sensitivity of the molecular assay for CK-detection or due to the Ficoll density gradient’s low-recovery yield of CTCs [65]. However, previous studies in metastatic breast cancer have shown that positive detection of CTCs using either RT-qPCR or IF is correlated with a notably decreased median overall survival [52]. According to our results, the agreement regarding CTC-positivity between these three different assays ranged from 48.0% to 68.0%. This observation aligns with previous studies highlighting the heterogeneity of CTCs, which contributes to the variability in results obtained from different isolation and detection methods. While these methods may not yield identical findings, they often provide complementary insights into CTC biology and characteristics [37]. Nevertheless, the combined and overlapping results between the two different detection methods resulted in a higher percentage of positivity for the examined biomarkers—*CXCR4*, *JUNB,* and *PDL1*. 

In summary, we investigated the expression of CXCR4, JUNB, and PD-L1 at the protein and molecular level in CTCs from mPCa patients. The protein-level study included EpCAM-independent isolation methods and two triple IF methods, while the molecular-level study included an EpCAM-dependent method and three individual RT-qPCR assays. The comprehensive analysis of CTCs using two different isolation and detection systems showed a high probability of positive CTC detection. Furthermore, the (CK+/CXCR4+/JUNB-) phenotype was associated with disease progression, and PD-L1 protein expression was also related to poorer OS for mPCa patients. This is a pilot study, as the patient number is limited; further studies with larger patient groups, encompassing different time points, could better clarify the above results. However, the study can provide insightful indications regarding the importance of these biomarkers for monitoring patient survival and possible therapeutic targets. Clinical trials for immunotherapies against the PD-1/PD-L1 axis have been tested in mPCa patients [3,66], while a combination of CXCR4-targeted therapy together with radiotherapy was more effective, inhibiting metastatic growth [67]. As long as a combination of PD-L1- and CXCR4-targeted therapies have been tested for TNBC patients [64], a relevant treatment combination could be tested in prostate cancer patients to examine their therapeutic impact.

One limitation of this study is the relatively small sample size, as well as the variability in the number of patients analyzed using different methods, such as the Ficoll density gradient, ISET platform, and RT-qPCR. The reason for these differences was the limited available blood volume from some patients. Future studies should aim to enroll a larger number of patients for each isolation technique and to analyze CXCR4, JUNB, and PD-L1 expression in a larger cohort of patients, with long-term follow-up to gain deeper insights and a more comprehensive understanding of their clinical relevance.

## 5. Conclusions

The present study examined CXCR4, JUNB, and PD-L1 at molecular and protein levels in circulating tumor cells (CTCs) derived from patients with mPCa. These biomarkers are known to play significant roles in cancer progression and metastasis. Our results showed complimentary overexpression of these biomarkers in CTCs of prostate cancer at both mRNA and protein levels. Meanwhile, the single-cell analysis revealed that the presence of the (CK+/CXCR4+/JUNB-) and (CK+/PD-L1+/CD45) phenotypes were correlated with poorer PFS and OS, respectively, providing interesting biomarkers for this type of cancer.

## Figures and Tables

**Figure 1 cells-13-00782-f001:**
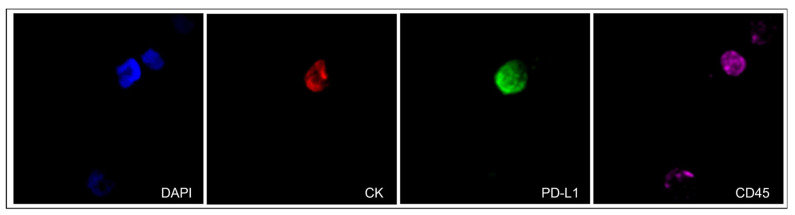
Expression of cytokeratin (CK) (red), PD-L1 (green), and CD45 (purple) in a CTC isolated from a metastatic prostate cancer (mPCa) patient (40× magnification).

**Figure 2 cells-13-00782-f002:**
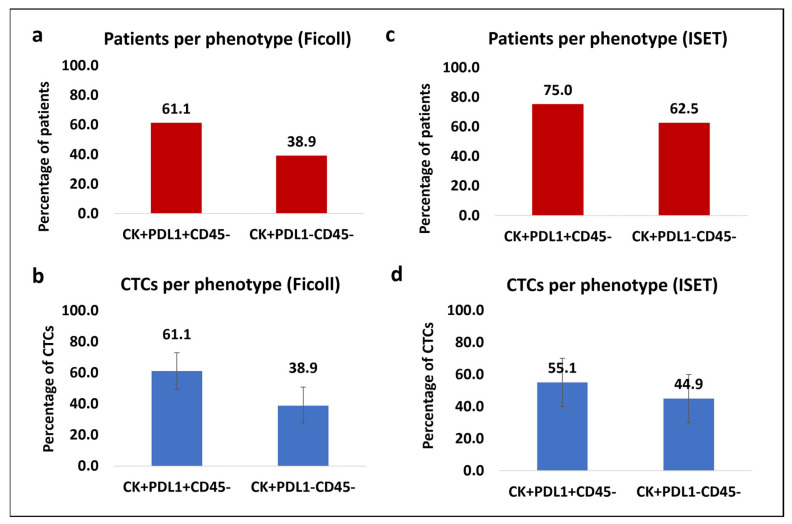
Phenotypic patterns of (CK/PD-L1/CD45) expression in mPCa blood samples. (**a**) Percentage of patients from Ficoll density gradient isolation with (CK+/PD-L1+/CD45-) and (CK+/PD-L1-/CD45-) phenotypes. (**b**) Average percentage of total CTCs from Ficoll density gradient isolation with (CK+/PD-L1+/CD45-) and (CK+/PD-L1-/CD45-) phenotypes. (**c**) Percentage of patients from ISET isolation with (CK+/PD-L1+/CD45-) and (CK+/PD-L1-/CD45-) phenotypes. (**d**) Average percentage of total CTCs from ISET isolation with (CK+/PD-L1+/CD45-) and (CK+/PD-L1-/CD45-) phenotypes.

**Figure 3 cells-13-00782-f003:**
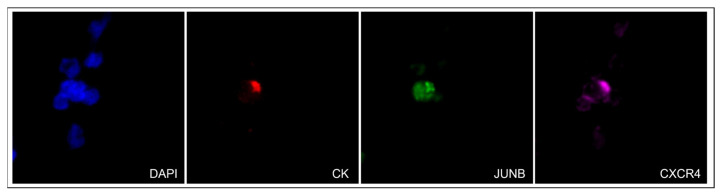
Expression of CK (red), JUNB (green), and CXCR4 (purple) in a CTC isolated from a patient with mPCa (40× magnification).

**Figure 4 cells-13-00782-f004:**
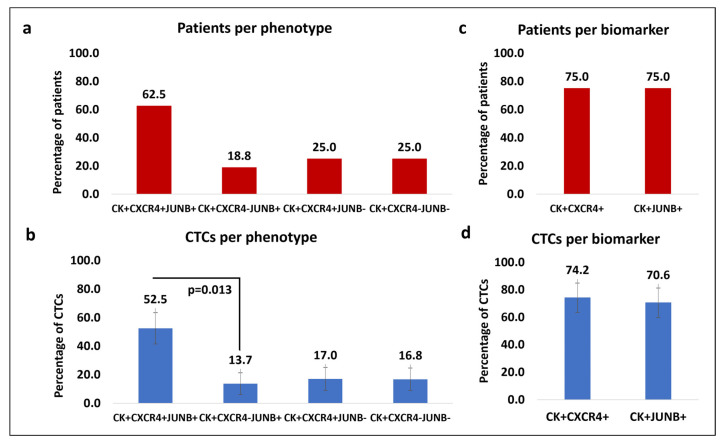
Phenotypic patterns of (CK/CXCR4/JUNB) expression in mPCa patients. (**a**) Percentage of patients with (CK+/CXCR4+/JUNB+), (CK+/CXCR4-/JUNB+), (CK+/CXCR4+/JUNB-), and (CK+/CXCR4-/JUNB-) phenotypes. (**b**) Average percentage of total CTCs with the identified phenotypes (CK+/CXCR4+/JUNB+), (CK+/CXCR4-/JUNB+), (CK+/CXCR4+/JUNB-), and (CK+/CXCR4-/JUNB-). (**c**) Percentage of patients with (CK+/CXCR4+) and (CK+/JUNB+) phenotypes. (**d**) Average percentage of total CTCs with (CK+/CXCR4+) and (CK+/JUNB+) phenotypes.

**Figure 5 cells-13-00782-f005:**
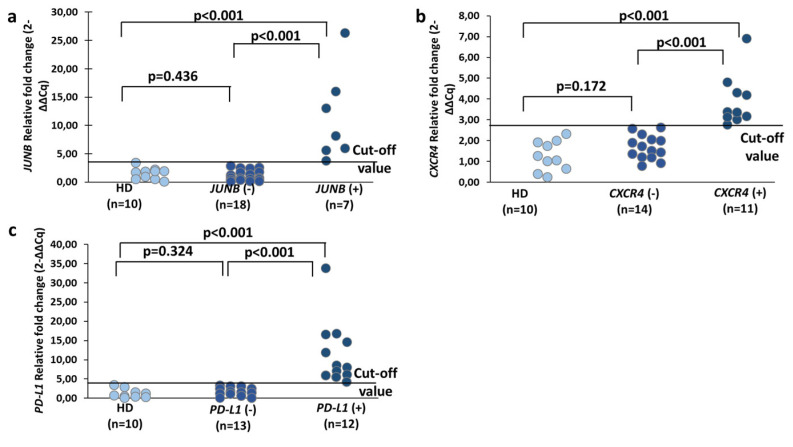
Relative fold-change values (2^−ΔΔCq^) for (**a**) *JUNB*, (**b**) *CXCR4*, (**c**) *PD-L1* in healthy individuals and CTC samples from mPCa patients.

**Figure 6 cells-13-00782-f006:**
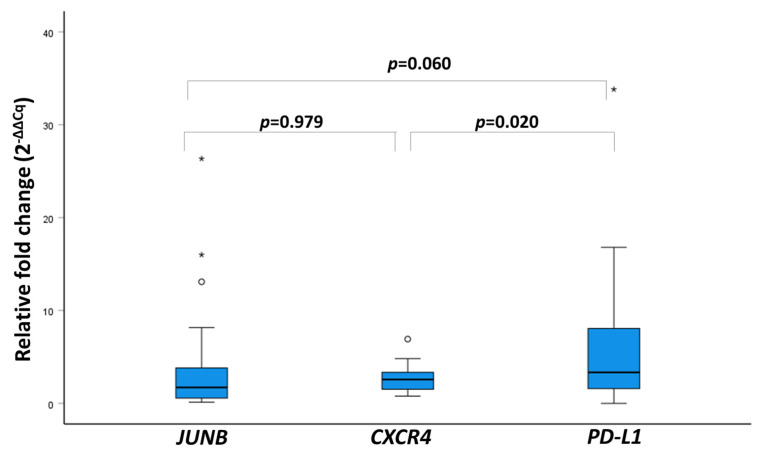
Boxplot representation and Wilcoxon signed-rank test results of relative fold-change values (2^−ΔΔCq^) for *JUNB*, *CXCR4*, and *PD-L1* from 25 mPCa patients (circles indicate potential outliers and asteriscs indicate extreme outliers).

**Figure 7 cells-13-00782-f007:**
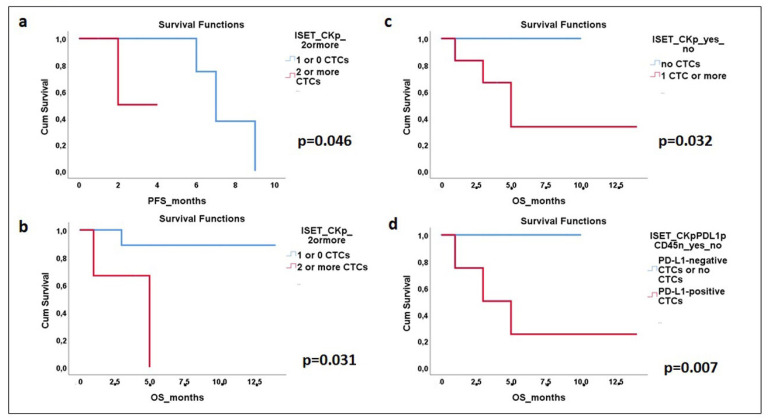
Kaplan—Meier survival curves in mPCa patients for CTCs and specific phenotypes, after ISET isolation. (**a**) Progression-free survival (PFS) for detection of ≥2 CTCs (HR: 1137.5, *p* = 0.046). (**b**) Overall survival (OS) for detection of ≥2 CTCs (HR: 9.0, *p* = 0.031). (**c**) OS for identification of ≥1 CTC (HR: 115.6, *p* = 0.032). (**d**) OS for the identification of (CK+/PD-L1+/CD45-) (HR: 262.7, *p* = 0.007).

## Data Availability

Data presented in the study are available upon request from the corresponding author.

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
