# Peer review of "Comprehensive Analysis of CXCR4, JUNB, and PD-L1 Expression in Circulating Tumor Cells (CTCs) from Prostate Cancer Patients"

_cells, 2024, doi:10.3390/cells13090782_

Round 1

Reviewer 1 Report

Comments and Suggestions for Authors

In this study, Roumeliotou et al. analysed the expression of CXCR4, JUNB, and PD-L1 in circulating tumour cells (CTCs) isolated from the blood of patients with metastatic prostate cancer (mPCa) to explore its association with clinical outcomes. While the methodological techniques are well-assessed with appropriate controls, and the manuscript is well-written, several major comments should be addressed for this study to be considered for publication in "Cells."

The primary concern lies in the lack of specificity regarding the selection criteria for the small subpopulations analysed. How were the 17 patients selected for ISET technology application? What criteria guided the selection process? Similarly, how were the 25 mPCa patients and 10 healthy individuals chosen for mRNA analysis? What are the demographic and clinical characteristics of these cohorts, and were there any significant differences in age or other relevant features?

Furthermore, it is essential to ensure consistency in RNA input across samples and investigate any variations in the expression levels of the housekeeping gene (B2M) among different groups. The validation of antibodies used for immunofluorescence (IF) should be thoroughly demonstrated, including evidence of absence of signal in siRNA-treated samples or negative controls.

Regarding PD-L1 expression, the study notes the absence of simultaneous PD-L1positive and PD-L1-negative CTCs in all patients. However, given the known focal nature of PD-L1 expression in prostate cancer, it is crucial to discuss whether this indicates homogeneous PD-L1 expression within these tumours or if it reflects a limitation in detecting PD-L1 heterogeneity in this study.

The authors should also disclose in detail the number of CTCs isolated per sample and address potential biases arising from variations in CTC numbers. Moreover, presenting the clinical information of patients, including Gleason score, number of metastases, metastasis site, tumour burden, etc., would be valuable for assessing any correlations with CTC number or survival analysis.

Finally, a minor correction is needed in the Introduction where "DTC" should be replaced with "CTC" (Line 94).

Reviewer 2 Report

Comments and Suggestions for Authors

The study delves into the expression and implications of CXCR4, JUNB, and PD-L1 biomarkers in circulating tumor cells (CTCs) isolated from patients with metastatic prostate cancer (mPCa). Utilizing Ficoll density gradient and ISET system for CTC isolation from 48 mPCa patients, the research employs triple immunofluorescence staining to identify specific CTC phenotypes and molecular analysis for gene expression profiling. The findings underscore the significant presence of CK+/PD-L1+/CD45– and CK+/CXCR4+/JUNB+ phenotypes, highlighting the association of CXCR4 expression with poorer progression-free survival and PD-L1 expression with decreased overall survival. Although hampered by small patient numbers the study presents an interesting analysis of CTC biomarkers, offering valuable insights into their role in mPCa progression and metastasis, and suggesting their potential as clinical biomarkers for prognosis and therapy.

Major Revisions

:Clarification of Results Presentation: The presentation of results, particularly the term "average percentage of total identified CTCs per patient," is confusing. For instance, the discrepancy between the calculated percentage (59.4% for 19 out of 32 CTCs, as seen in Table S2) and the reported 61.1% in line 256 and Figure 2b needs clarification. Similar issues are observed with the percentages in line 278 and Figure 2d. Please ensure that the calculations and presentations of percentages are consistent and clearly explained.

Reporting of CTC Counts: Given the low total number of CTCs per patient, it is crucial to report the average CTC count, including the range, in the main text to emphasize the significance of these findings.

Patient Follow-up Data: Please provide the median follow-up duration for patients in Section 3.4 to enhance the understanding of the study's longitudinal aspects.

Group Classification Clarity: The classification of groups in Section 3.3, as inferred from Figure S5a, appears ambiguous. Specifically, does CK+JUNB+ indicate the presence of either CK+ or JUNB+ markers, or both? Clarification is needed for better interpretation of the results.

Consistency in Patient ID References: There appears to be a discrepancy between patient IDs in Figure S5a and those listed in Tables S2, S4, and S5. Aligning these references could enrich the discussion by linking gene expression profiles to CTC counts more effectively.

Heterogeneity of CTCs: Figure S5a reveals the presence of CK- cells that are positive for JUNB, CXCR4, or PD-L1, suggesting significant heterogeneity among CTCs. This critical observation about the limitations of epithelial marker-based CTC enrichment is not discussed in the manuscript and warrants attention.

Comparison of Marker Expression: The data presented in Figure S5a and Figure 6 seem to contradict each other regarding the expression of CXCR4 and JUNB. Specifically, Figure S5a shows more CXCR4-overexpressed but JUNB-negative samples compared to the reverse, which is inconsistent with the impression given in Figure 6. This discrepancy should be addressed for a coherent understanding of marker expression dynamics.

Minor Revisions:

Terminology Consistency: In line 405, the term "method" should be used instead of "methods" for consistency and accuracy

.Figure Legibility: The font size on the axes in Figure 7 is too small, potentially hindering readability. Additionally, please verify whether there are indeed no censored events in Figures 7a-d as indicated.

Consistency of Statistical Data: The p-values reported in line 338 do not match those shown in Figure 5. Ensuring consistency between textual and visual data presentations is essential for credibility.

Isolation Method Observations: The manuscript notes more CK+ RT-qPCR results from Epcam-based isolation compared to size-based enrichment and staining, as seen in Figure S5a. This intriguing finding deserves further discussion, possibly including speculation on methodological differences or implications for CTC detection efficacy.

Reviewer 3 Report

Comments and Suggestions for Authors

This study was reported the useful predictors of CXCR4, JUNB, and PD-L1 expression in CTCs for metastatic prostate cancer. The reviewer would like to suggest some critiques as follows.

1.      A fundamental problem with this study is the small number of cases that tested positive for each factor; the Limitation should be described in detail. Although the difference between patients with mCRPC and healthy volanteers is not mentioned, it should be described what the difference is.

2.      “metastatic prostate cancer patients” is not so good. “patients with prostate cancer who had distant metastases” is better. The authors should revise this point.

3.      On line 33 and 34, PFS and OS are listed in the wrong place.

4.      On line 65, “Certain CTC … therapeutic response” is unclear. The endpoints in this paper do not appear to be related to treatment efficacy.

5.      On line 76, what is HNSCC?

6.      On line 76, “at the end of definitive treatment” is unclear. RT? Surgery? Systemic therapy?

7.      On line 94, what is DTCs?

8.      On line 97, what is poorer survival? PFS? OS? Oncological outcomes?

9.      Are there any differences between treatment-naive patients and patients with prostate cancer after completion of first-line treatment?

10.  The positive rate of CTC is different in lines 253 and 294, is this statement correct?

11.  The number of patients who tested positive for each factor is small, but the reader may perceive the number as large in percentage terms. The method of description needs to be devised.

12.  In Figure 7, the number at risk should be listed. The median PFS and OS should be listed for each group.

13.  On line 426, the relationship between this sentence and the purpose of this study is not clear.

Round 2

Reviewer 1 Report

Comments and Suggestions for Authors

thank you for all the corrections and explanation that have improved the research and reproducibility.

Reviewer 2 Report

Comments and Suggestions for Authors

I am content with the additions and changes to the manuscript as well as with the comments to my suggestions

Reviewer 3 Report

Comments and Suggestions for Authors

None.